# SARS-CoV-2 is transmitted via contact and via the air between ferrets

Mathilde Richard [1], Adinda Kok [1], Dennis de Meulder[1], Theo M. Bestebroer[1], Mart M. Lamers [1], Nisreen M. A. Okba [1], Martje Fentener van Vlissingen [2], Barry Rockx [1], Bart L. Haagmans [1], Marion P. G. Koopmans [1], Ron A. M. Fouchier [1] & Sander Herfst [1✉]

SARS-CoV-2, a coronavirus that emerged in late 2019, has spread rapidly worldwide, and information about the modes of transmission of SARS-CoV-2 among humans is critical to apply appropriate infection control measures and to slow its spread. Here we show that SARS-CoV-2 is transmitted efficiently via direct contact and via the air (via respiratory droplets and/or aerosols) between ferrets, 1 to 3 days and 3 to 7 days after exposure respectively. The pattern of virus shedding in the direct contact and indirect recipient ferrets is similar to that of the inoculated ferrets and infectious virus is isolated from all positive animals, showing that ferrets are productively infected via either route. This study provides experimental evidence of robust transmission of SARS-CoV-2 via the air, supporting the implementation of community-level social distancing measures currently applied in many countries in the world and informing decisions on infection control measures in healthcare settings.

[1] Department of Viroscience, Erasmus University Medical Center, Rotterdam, The Netherlands. [2] Erasmus Laboratory Animal Science Center, Erasmus University Medical Center, Rotterdam, The Netherlands. ✉email: s.herfst@erasmusmc.nl

In late December 2019, clusters of patients in China presenting with pneumonia of unknown etiology were reported to the World Health Organization (WHO)[1]. The causative agent was rapidly identified as being a virus from the *Coronaviridae* family, closely related to the severe acute respiratory syndrome coronavirus (SARS-CoV)[2–4]. The SARS-CoV epidemic affected 26 countries and resulted in more than 8000 cases in 2003. The newly emerging coronavirus, named SARS-CoV-2[5], rapidly spread worldwide and was declared pandemic by the WHO on March 11, 2020[6]. The first evidence suggesting human-to-human transmission came from the descriptions of clusters among the early cases[7,8]. Based on epidemiological data from China before measures were taken to control the spread of the virus, the reproductive number R0 (the number of secondary cases directly generated from each case) was estimated to be between 2 and 3[9–11]. In order to apply appropriate infection control measures to reduce the R0, the modes of transmission of SARS-CoV-2 need to be elucidated. Respiratory viruses can be transmitted via direct and indirect contact (via fomites), and through the air via respiratory droplets and/or aerosols. Transmission via respiratory droplets (>5 μm) is mediated by expelled particles that have a propensity to settle quickly and is therefore reliant on close proximity between infected and susceptible individuals, usually within 1 m of the site of expulsion. Transmission via aerosols (<5 μm) is mediated by expelled particles that are smaller in size than respiratory droplets and can remain suspended in the air for prolonged periods of time, allowing infection of susceptible individuals at a greater distance from the site of expulsion[12]. Current epidemiological data suggest that SARS-CoV-2 is transmitted primarily via respiratory droplets and contact[7–9,13,14], which is used as the basis for mitigation of spread through physical and social distancing measures. However, scientific evidence that SARS-CoV-2 can be efficiently transmitted via the air is weak.

Previous studies have shown that ferrets were susceptible to infection with SARS-CoV[15–19], and that SARS-CoV was efficiently transmitted to co-housed ferrets via direct contact[15]. Here, we use a ferret transmission model to show that SARS-CoV-2 spreads through direct contact and through the air (via respiratory droplets and/or aerosols).

## Results
**Transmission of SARS-CoV-2 between ferrets.** Individually housed donor ferrets were inoculated intranasally with a strain of SARS-CoV-2 isolated from a German traveller returning from China. Six hours post-inoculation (hpi), a direct contact ferret was added to each of the cages. The next day, indirect recipient ferrets were placed in adjacent cages, separated from the donor cages by two steel grids, 10 cm apart, allowing viruses to be transmitted only via the air (Fig. 1). On alternating days to prevent cross-contamination, throat, nasal and rectal swabs were collected from each ferret in the inoculated and direct contact groups and from the indirect recipient group, followed by SARS-CoV-2 detection by RT-qPCR and virus titration.

Ferrets were productively infected by SARS-CoV-2 upon intranasal inoculation, as demonstrated by the robust and long-term virus shedding from the donor ferrets (Fig. 2, Supplementary Fig. 1). SARS-CoV-2 RNA levels peaked at 3 days post-inoculation (dpi) and were detected up to 11 dpi in two animals and up to 15 and 19 dpi in the other two animals (Fig. 2, Supplementary Fig. 1). SARS-CoV-2 was transmitted to direct contact ferrets in four out of four independent experiments between 1 and 3 days post-exposure (dpe) and viral RNA was detected up to 13–15 days (i.e. 13–17 dpe) (Fig. 2, Supplementary Fig. 1). Interestingly, SARS-CoV-2 was also transmitted via the air to three out of four indirect

recipient ferrets. SARS-CoV-2 RNA was detected from 3 to 7 dpe onwards these indirect recipient ferrets and for 13–19 days (Fig. 2, Supplementary Fig. 1).

Whereas donor ferrets were inoculated with a high virus dose, direct contact and indirect recipient ferrets are likely to have received a low infectious dose via direct contact or via the air. In spite of this, the pattern of virus shedding from the direct contact and indirect recipient ferrets was similar to that of the inoculated donor ferrets, both in terms of duration and SARS-CoV-2 RNA levels, corroborating robust replication of SARS-CoV-2 upon transmission via direct contact and via the air, independent of the infectious dose. In general, higher SARS-CoV-2 RNA levels were detected in the throat swabs as compared to the nasal swabs. SARS-CoV-2 RNA levels in the rectal swabs were overall the lowest. From each SARS-CoV-2 RNA positive animal, infectious virus was isolated in VeroE6 cells from throat and nasal swabs for at least two consecutive days (Supplementary Fig. 2 and Supplementary Table 1). In contrast, no infectious virus was isolated from the rectal swabs. Infectious virus titers ranged from $10^{0.75}$ to $10^{2.75}$ TCID$_{50}$/ml (median tissue culture infectious dose per ml) in the donor ferrets, from $10^{0.75}$ to $10^{3.5}$ TCID$_{50}$/ml in the direct contact ferrets and from $10^{0.75}$ to $10^{4.25}$ TCID$_{50}$/ml in the indirect recipient ferrets. All SARS-CoV-2 positive ferrets seroconverted 21 dpi/dpe, and the antibody levels detected using a receptor binding domain (RBD) enzyme-linked immunosorbent assay (ELISA) were similar in donor, direct contact and indirect recipient ferrets (Fig. 3a). Plaque reduction neutralization titers (PRNT) in sera from indirect recipient ferrets were lower than that of donor and direct contact ferrets, which is probably due to the later onset of virus replication upon transmission via the air and thus a relatively earlier collection of serum after infection (Fig. 3b). The indirect recipient ferret, in which no SARS-CoV-2 was detected, did not seroconvert as expected.

**Sequence analysis of viruses isolated from ferrets.** MinION (Nanopore) sequencing was used to determine the whole genome consensus sequences of viruses in throat swabs collected from the four donor (all 3 dpi), the four direct contact (all 5 dpe) and three indirect recipient ferrets (7, 9 or 11 dpe). Two substitutions were detected in the consensus sequence of viruses collected from all ferrets as compared to the sequence of the original virus isolate: N501T and S686G, both in the spike protein. Residue 501 is part of the receptor binding motif that mediates contact with angiotensin-converting enzyme 2 (ACE2), the receptor of SARS-CoV-2. A threonine at position 501 (as present in the majority of SARS-CoV viruses) was previously shown to decrease the affinity of the spike protein with the receptor[20]. Perhaps this substitution emerged in ferrets as a result of adaptation to efficient binding to ferret ACE2. Residue 686 is the first residue after the furin cleavage site. The serine to glycine substitution has not been found in human SARS-CoV-2 sequences, hence the effect of this substitution is unknown. In addition, a L260F substitution in Nsp6 was observed in the throat swab of a direct contact ferret, and two synonymous substitutions (C2910T and C7235T) were detected in an indirect recipient ferret and a direct contact ferret respectively. In order to understand whether the N501T and S686G substitutions were either positively selected in ferrets from existing minority variants in the virus isolate or had mutated in ferrets, Illumina next-generation sequencing was performed on sequential samples from the donor ferrets and on the virus stocks (Supplementary Table 2). Single nucleotide polymorphisms (SNPs) which were present in >5% of the total number of reads were called (Supplementary Table 2A). The N501T substitution was already present in all donor ferrets at 1 dpi in 14.7%–49.6% of the reads and percentages rapidly increased to 86.4%–98,7% on

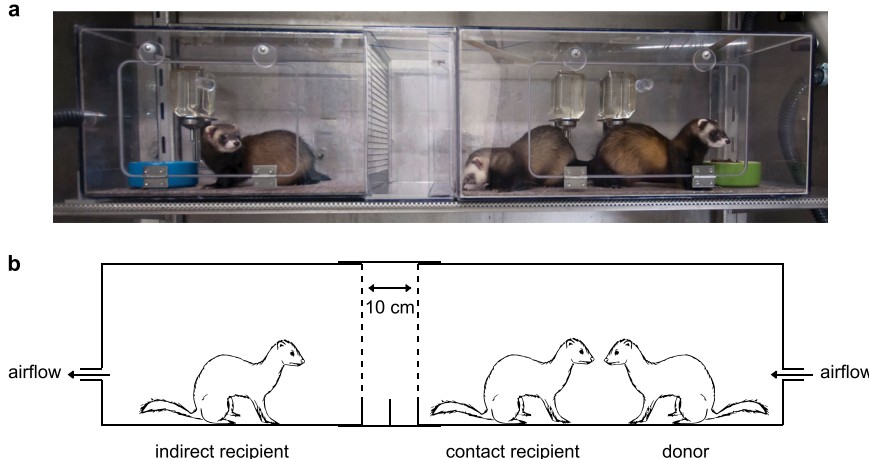

**Fig. 1 The ferret transmission experimental set-up.** Picture (**a**) and schematic representation (**b**) of one independent experimental set-up to assess direct contact transmission and indirect transmission via the air. One inoculated donor ferret is housed in a cage (right-hand side of the picture). Six hours later, a direct contact ferret is added to the same cage as the donor ferret. The next day, an indirect recipient ferret is placed in an opposite cage (left-hand side of the picture) separated by two steel grids, 10 cm apart, to avoid contact transmission. The direction of the air flow (100 L min$^{-1}$) is indicated by the arrows. The ferret transmission set-ups are placed in class III isolators in a biosafety level 3+ laboratory.

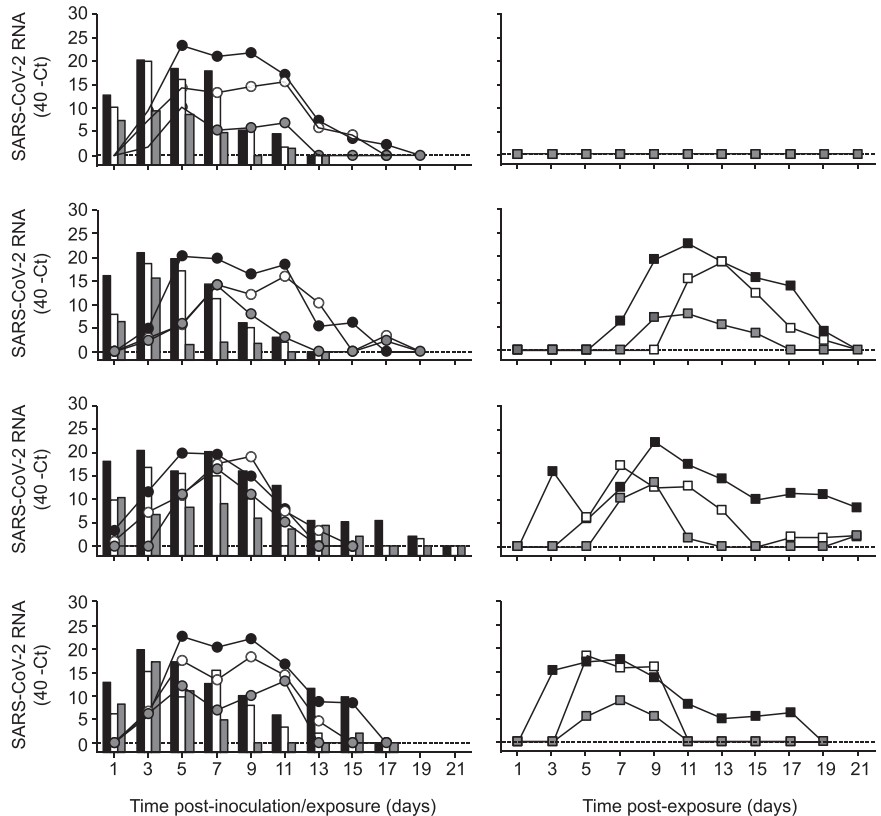

**Fig. 2 SARS-CoV-2 shedding in ferrets in the transmission experiment.** SARS-CoV-2 viral RNA was detected by RT-qPCR in throat (black), nasal (white) and rectal (grey) swabs collected from inoculated donor ferrets (bars; left panels), direct contact ferrets (circles; left panels) and indirect recipient ferrets housed in separate cages (squares; right panels). Swabs were collected from each ferret every other day until no viral RNA was detected in any of the three swabs. The dotted line indicates the detection limit.

3 dpi. At 7 dpi, the percentages of reads with the N501T substitution were still high, albeit lower in donor ferret 4 (66.2%). A similar trend was observed for the S686G substitution. In addition, an R685H substitution in the spike protein was detected in two ferrets and L207F and L260F substitutions in Nsp6 were detected in individual ferrets, all at low percentages (Supplementary Table 2A). SNP analysis of the virus isolates

demonstrated that S686G was the only substitution that was present in more than 5% of the reads: 8.1% in the passage 3 virus stock used to inoculate donor ferrets and 15,2% in the passage 1 virus isolate from Germany (Supplementary Table 2B). Among the other substitutions observed in the ferret samples, only the R685H substitution was detected at >1% of the reads in the original virus isolate (Supplementary Table 2B).

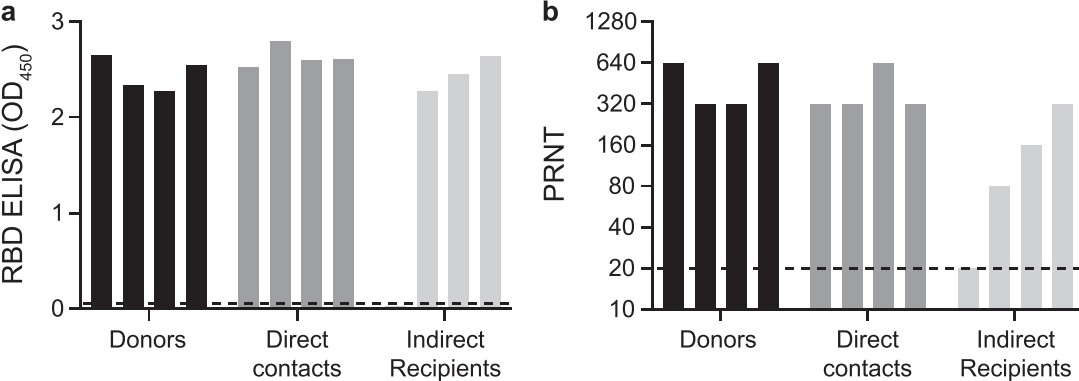

**Fig. 3 Antibody responses in donor, direct contact and indirect recipient ferrets 21 dpi/dpe.** Sera were collected from the donor, direct contact and indirect recipient ferrets 21 dpi/dpe and IgG responses were assessed using a SARS-CoV-2 receptor binding site (RBD) ELISA (**a**) and using a plaque reduction neutralization assay (**b**). The dotted lines indicate the detection limit of the assays. PRNT: plaque reduction neutralization titer. OD: optic density. All presera were tested negative by RBD ELISA and plaque reduction neutralization assay (OD$_{450}$ 0.02–0.05; PRNT < 20).

## Discussion

Here, we show that SARS-CoV-2 is transmitted via contact and via the air between ferrets. SARS-CoV-2 transmission in experimental animal models has recently also been described by others. SARS-CoV-2 direct contact transmission between ferrets[21] and hamsters[22] was reported, with similar efficiency as observed in our study. In addition, SARS-CoV-2 was also found to be transmitted via the air in two out of six ferrets[21], and in two out of six cats[23]. However, only low levels of SARS-CoV-2 RNA were detected in nasal washes and feces of the indirect recipient ferrets, and no infectious virus was isolated[21]. Furthermore, virus shedding was shorter as compared to the donor animals and only one out of the two SARS-CoV-2 RNA positive indirect recipient ferrets seroconverted. Similarly, the transmission via the air between cats was not efficient. SARS-CoV-2 RNA was detected in the feces and tissues of one cat at 3 and 11 dpi, respectively and in nasal washes of another cat, but no infectious virus was isolated. Both SARS-CoV-2 RNA positive indirect recipient cats seroconverted. In contrast, the present study showed that SARS-CoV-2 was efficiently transmitted via the air between ferrets, as demonstrated by long-term virus shedding and the presence of infectious virus in the indirect recipient animals, which is comparable to the transmissibility of pandemic influenza viruses in the ferret model[24].

To date, there is no evidence of fecal-oral transmission of SARS-CoV-2 in humans. However, the prolonged detection of RNA in consecutive stool samples[25] and the environmental contamination of sanitary equipment[26] may suggest that the fecal-oral route could be a potential route of transmission of SARS-CoV-2. Here, no infectious virus was retrieved from any of the rectal swabs. Despite this, it cannot be fully excluded that SARS-CoV-2 was also transmitted from donors to direct contact ferrets partly via the fecal-oral route. In the study by Kim et al., ferret fecal material was used to inoculate ferrets, resulting in a productive infection, indicating that infectious SARS-CoV-2 was shed in fecal specimens[21].

Our experimental system does not allow to assess whether SARS-CoV-2 was transmitted via the air through respiratory droplets, aerosols or both, as donor and indirect recipient ferret cages are placed only 10 cm apart from each other. In a recent study, SARS-CoV-2 remained infectious in aerosols for at least 3 h after aerosolization at high titers in a rotating drum, comparable to SARS-CoV[27]. Although it is informative to compare the stability of different respiratory viruses in the air, our study provides the additional information that infectious SARS-CoV-2 particles can actually be expelled in the air and subsequently infect recipients. In two other studies, the presence of SARS-CoV-2 in air samples collected in hospital settings was investigated. However, no SARS-CoV-2 RNA was detected in the air sampled in three isolation rooms[26], or 10 cm from a symptomatic patient who was breathing, coughing or speaking[28]. Nevertheless, RNA was detected on the air exhaust outlet of one of the isolation rooms in the first study, suggesting that virus-laden droplets may be displaced by airflows[26].

Here we provide the first experimental evidence that SARS-CoV-2 can be transmitted efficiently via the air between ferrets, resulting in a productive infection and the detection of infectious virus in indirect recipients, as a model for human-to-human transmission. Although additional experiments on the relative contribution of respiratory droplets and aerosols to the transmission of SARS-CoV-2 are warranted, the results of this study corroborate the WHO recommendations about transmission precautions in health care settings and the social distancing measures implemented in many countries around the globe to mitigate the spread[29]. The ferret transmission model will also be useful to understand transmission dynamics and the molecular basis of the transmissibility of SARS-Cov-2 and other betacoronaviruses, which, in the context of the current SARS-CoV-2 pandemic and future pandemic threats, is clearly of utmost importance.

## Methods

**Virus and cells**. SARS-CoV-2 (isolate BetaCoV/Munich/BavPat1/2020; GISAID ID EPI_ISL 406862; kindly provided by Prof. Dr. C. Drosten) was propagated to passage 3 on VeroE6 cells (ATCC) in Opti-MEM I (1×) + GlutaMAX (Gibco), supplemented with penicillin (10,000 IU mL$^{-1}$, Lonza) and streptomycin (10,000 IU mL$^{-1}$, Lonza) at 37 °C in a humidified CO2 incubator. VeroE6 cells were inoculated at an moi of 0.01. Supernatant was harvested 72 hpi, cleared by centrifugation and stored at –80 °C. The virus stock was tested mycoplasma negative and contained 3.15 × 10$^8$ genome copies/ml (RdRp gene).

VeroE6 cells were maintained in Dulbecco modified Eagle medium (DMEM, Gibco) supplemented with 10% foetal calf serum (Greiner), 2 mM of L-glutamine (Gibco), 10 mM Hepes (Lonza), 1.5 mg ml$^{-1}$ sodium bicarbonate (NaHCO$_3$, Lonza), penicillin (10,000 IU/mL) and streptomycin (10,000 IU/mL) at 37 °C in a humidified CO$_2$ incubator. All work was performed in a Class II Biosafety Cabinet under BSL-3 conditions at the Erasmus Medical Center.

**Ferret transmission experiment**. All relevant ethical regulations for animal testing have been complied with. Animals were housed and experiments were performed in strict compliance with the Dutch legislation for the protection of animals used for scientific purposes (2014, implementing EU Directive 2010/63). Influenza virus, SARS-CoV-2 and Aleutian Disease Virus seronegative 6 month-old female ferrets (*Mustela putorius furo*), weighing 700–1000 g, were obtained from a commercial breeder (TripleF (USA)). Research was conducted under a project license from the Dutch competent authority (license number AVD1010020174312) and the study protocol was approved by the institutional Animal Welfare Body

(Erasmus MC permit number 17-4312-02). Animal welfare was monitored on a daily basis. Virus inoculation of ferrets was performed under anesthesia with a mixture of ketamine/medetomidine (10 and 0.05 mg kg$^{-1}$, respectively) antagonized by atipamezole (0.25 mg kg$^{-1}$). Swabs were taken under light anesthesia using ketamine to minimize animal discomfort.

Four donor ferrets were inoculated intranasally with $6 \times 10^5$ TCID$_{50}$ of SARS-CoV-2 virus diluted in 500 µl of phosphate-buffered saline (PBS) (250 µl instilled dropwise in each nostril) and were housed individually in a cage. Six hpi, direct contact ferrets were placed in the same cage as the donor ferrets. One day later, indirect recipient ferrets were placed in an opposite cage separated by two steel grids, 10 cm apart, to avoid contact transmission (Supplementary Fig. S1). The air flow rate from the donor to the recipient ferret was ~100 L min$^{-1}$ and the temperature of the room was between 21 and 22 °C. Throat, nasal and rectal swabs were collected using dry swabs (Coban, cat. 155CS01) every other day, to prevent cross-contamination, until they were negative for SARS-CoV-2 RNA or maximum for 21 dpi/dpe by determined by real-time RT-qPCR as described below. Swabs were stored at −80 °C in transport medium (Minimum Essential Medium Eagle with Hank's BSS (Lonza), 5 g L$^{-1}$ lactalbumine enzymatic hydrolysate (Sigma-Aldrich), 10% glycerol (Sigma-Aldrich), 200 U ml$^{-1}$ of penicillin, 200 mg ml$^{-1}$ of streptomycin, 100 U ml$^{-1}$ of polymyxin B sulfate (Sigma-Aldrich), and 250 mg ml$^{-1}$ of gentamicin (Life Technologies)) for end-point titration in VeroE6 cells as described below. Ferrets were euthanized at 21 dpi/dpe by heart puncture under anaesthesia. Therefore, the exposure duration of direct contact and indirect recipient ferrets was 21 and 20 days respectively. Blood was collected in serum-separating tubes (Greiner) and processed according to the manufacturer's instructions. Sera were heated for 1 h at 60 °C and used for the detection of specific antibodies against SARS-CoV-2 as described below. All animal experiments were performed in class III isolators in a negatively pressurized ABSL3+ facility.

**RNA isolation and RT-qPCR**. RNA was isolated using an in-housed developed high-throughput method in a 96-well format. Sixty µl of sample were added to 90 µl of MagNA Pure 96 External Lysis Buffer (Roche). A known concentration of phocine distemper virus (PDV) was added to the sample as internal control for the RNA extraction[30]. The 150 µl of sample/lysis buffer was added to a well of a 96-well plate containing 50 µl of magnetic beads (AMPure XP, Beckman Coulter). After thorough mixing by pipetting up and down at least 10 times, the plate was incubated for 15 minutes (min) at room temperature. The plate was then placed on a magnetic block (DynaMag™-96 Side Skirted Magnet (ThermoFisher Scientific)) and incubated for 3 min to allow the displacement of the beads towards the side of the magnet. Supernatants were carefully removed without touching the beads and beads were washed three times for 30 seconds (sec) at room temperature with 200 µl/well of 70% ethanol. After the last wash, a 10 µl multi-channel pipet was used to remove residual ethanol. Plates were air-dried for 6 min at room temperature. Plates were removed from the magnetic block and 30 µl of PCR grade water was added to each well and mixed by pipetting up and down 10 times. Plates were incubated for 5 min at room temperature and then placed back on the magnetic block for 2 min to allow separation of the beads. Supernatants were pipetted in a new plate and RNA was kept at 4 °C. Eight µl of RNA were directly pipetted into a mix for RT-qPCR, containing 0.4 µl of primers and probe mix targeting the E gene of SARS-CoV-2 (forward primer: 5′-ACAGGTACGTTAATAGTTAATAGCGT-3′; reverse primer: 5′-ATATTGCAGC AGTACGCACACA-3′; probe: 5′-FAM-ACACTAGCCATCCTTACTGCGCTTCG-B HQ-3′)[31], 0.4 µl of primers and probe mix targeting the HA gene of PDV (forward primer: 5′-CGGGTGCCTTTTACAAGAAC-3′; reverse primer: 5′-TTCTTTCCTCA ACCTCGTCC-3′, probe: 5′-Cy5-ATGCAAGGGCCAATTCTTCCAAGTT-BHQ-3′), 4 µl of TaqMan™ Fast Virus 1-Step Master Mix (ThermoFisher Scientific) and 6.2 µl of PCR grade water. Amplification and detection was performed on an ABI7700 (ThermoFischer Scientific) using the following program: 5 min 50 °C, 20″ 95 °C, [3″ 95 °C, 31″ 58 °C] × 45 cycles.

**Virus titrations**. Throat, nasal and rectal swabs were titrated in quadruplicates in VeroE6 cells. Briefly, confluent VeroE6 cells were inoculated with 10-fold serial dilutions of sample in Opti-MEM I (1×) + GlutaMAX, supplemented with penicillin (10,000 IU mL$^{-1}$), streptomycin (10,000 IU mL$^{-1}$). At one hpi, the first three dilutions were washed twice with media and fresh media was subsequently added to the whole plate. At six dpi, virus positivity was assessed by reading out cytopathic effects. Infectious virus titers (TCID$_{50}$/ml) were calculated from four replicates of each throat, nasal and rectal swabs and from 24 replicates of the virus stock using the Spearman–Karber method.

**Serology**. Pre-sera (collected before the start of the experiment) and sera collected at 21 dpi/dpe were tested for SARS-CoV-2 antibodies using a receptor binding domain (RBD) enzyme-linked immunosorbent assay (ELISA)[32]. ELISA plates were coated overnight at 4 °C with 100 ng/well of in-housed produced SARS-CoV-2 RBD diluted in PBS. After blocking with Blocker™ BLOTTO in TBS (Life technologies) + 0.01% of Tween-20 (Sigma-Aldrich), heat-inactivated sera (diluted 1:100) were added and incubated for 1 h at 37 °C. Bound antibodies were detected using horseradish peroxidase (HRP)-labelled goat anti-ferret IgG (1:10,000; ab112770, Abcam) and 3,3′,5,5′-Tetramethylbenzidine (TMB, Life Technologies) as a substrate. The absorbance of each sample was measured at 450 nm.

Additionally, presera and sera collected at 21 dpi/dpe were tested for the presence of SARS-CoV-2 neutralizing antibodies using a plaque reduction neutralization test (PRNT)[32]. Heat-inactivated sera were two-fold serially diluted in DMEM supplemented with NaHCO$_3$, HEPES buffer, penicillin, streptomycin, and 1% fetal bovine serum, starting at a dilution of 1:10 in 50 µL. Fifty µL of diluted virus suspension (400 plaque-forming units) were added and the mixture was incubated for 1 h at 37 °C. The mixtures were then placed on VeroE6 cells and incubated for 8 h. After incubation, cells were fixed with 4% formaldehyde/phosphate-buffered saline (PBS) and stained with a monoclonal mouse anti-SARS-CoV nucleocapsid antibody (1:10,000; 40143-MM05, Sino Biological), and a secondary HRP-labeled goat anti-mouse IgG1 (1:2000; 1071-05, Southern Biotech). HRP was revealed using the 3,3′,5,5′-tetramethylbenzidine substrate (True Blue; Kirkegaard and Perry Laboratories) and the number of infected cells per well was assessed by using ImmunoSpot Image Analyzer (CTL Europe GmbH). The plaque reduction neutralization titer (PRNT) was determined as the reciprocal of the highest dilution resulting in a reduction of >90% of the number of infected cells. The detection limit of the assay was <20.

**Whole genome sequencing using MinION**. A SARS-CoV-2 specific multiplex PCR was performed as recently described[33]. In short, primers for 86 overlapping amplicons spanning the entire genome were designed using primal scheme (http://primal.zibraproject.org/.) (for primer sequences, see Supplementary Table 3). The amplicon length was set to 500 bp with 75 bp overlap between the different amplicons. The libraries were generated using the native barcode kits from Nanopore (EXP-NBD104 and EXP-NBD114 and SQK-LSK109) and sequenced on a MinION R9.4 flow cell multiplexing up to 24 samples per sequence run according to the manufacturer's instructions.

The resulting raw sequence data were demultiplexed using Porechop (https://github.com/rrwick/Porechop). FASTQ files were then imported to the CLC Genomics Workbench v20.0.3 (QIAGEN) for analysis. First, sequences were trimmed off 33 base pairs on both the 3′ and 5′ ends to remove primer sequences and also using a Phred quality score threshold of 8. The trimmed sequences were mapped to the reference sequence (GISAID ID EPI_ISL 406862) with the following default parameters (match score = 1, mismatch cost = 2, insertion cost = 3, length fraction = 0.5 and similarity fraction = 8) and consensus genomes were extracted.

**Next-generation sequencing**. Amplicons were generated by a SARS-CoV-2 specific multiplex PCR as described above for the whole genome sequencing. Amplicons were purified with 0.8x AMPure XP beads (Beckman Coulter) and 100 ng of DNA was converted into paired-end Illumina sequencing libraries using KAPA HyperPlus library preparation kit (Roche), following the manufacturer's recommendations, to enable subsequent sequencing of multiple libraries in a single Illumina V3 MiSeq flowcell (2×300 cycles). Multiplex Adaptors (KAPA Unique Dual-Indexed Adapters Kit (Roche)) with indexes were used. FASTQ files were then imported to the CLC Genomics Workbench v20.0.3 (QIAGEN) for analysis. First, sequences were trimmed off 33 base pairs on both the 3′ and 5′ ends to remove primer sequences and also using Phred quality score threshold of 20. The trimmed sequences were mapped to the reference sequence (GISAID ID EPI_ISL 406862) with the following default parameters (match score = 1, mismatch cost = 2, insertion cost = 3, length fraction = 0.5 and similarity fraction = 8). Variants were called with the Basic Variant Detection tool. Single nucleotide polymorphisms that were present in both the forward and reverse reads with a 200x minimum coverage and a minimum variant count of 10 (5%) were called.

**Reporting summary**. Further information on research design is available in the Nature Research Reporting Summary linked to this article.

## Data availability

All data are available from the corresponding author (S.H.) on reasonable request. Porechop, which was used to demultiplex data from the MinION and Illumina sequencing, is available on github at: https://github.com/rrwick/Porechop. The source data underlying Figs. 2, 3, Supplementary Figs. 2 and 3 are provided as a Source data file. The sequencing raw data were deposited in the NCBI Sequence Read Archive (SRA) under the BioProject PRJNA641813. Source data are provided with this paper.

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

## Acknowledgements

We thank Prof. Dr. Christian Drosten (Charité—Universitätsmedizin Berlin) for providing the SARS-CoV-2 isolate used in this study and Drs Rik de Swart and Mathieu Sommers for their help with animal ethics and study approval and Dr. Bas Oude Munnink for providing the protocol of the Minion sequencing. This work was supported by European Union's Horizon 2020 research and innovation program VetBioNet (grant agreement No 731014) and NIH/NIAID (contract number HHSN272201400008C). S.H. was funded in part by an NWO VIDI grant (contract number 91715372).

## Author contributions

M.R. and S.H. conceived, designed, analysed and performed the work. M.R. and S.H. wrote the manuscript. A.K., D.M., T.B., M.L., and N.O. helped with performing the work. M.F.V., B.R., B.H., M.K., and R.A.M.F. helped with the design of the work, interpretation of the data and manuscript revision. All authors read and approved the final manuscript.

## Competing interests

The authors declare no competing interests.
