## [Peer Review File · Nature Communications]

Reviewers' Comments:

Reviewer #1:

Remarks to the Author:

The manuscript submission by Sander Herfst and colleagues reports on the direct and airborne transmission of SARS-CoV-2 in a ferret model of COVID-19. The authors demonstrate in this animal model that virus transmission is higher by direct contact versus an airborne route of transmission. Their findings support infection prevention measures, such as social distancing, to limit the public health impact. There are several minor concerns that should be addressed prior to acceptance for publication.

Item 1. The data presented in supplementary figure 2 should be presented as figure 2 as infectious virus titers are more relevant for public health risk assessment. The manuscript should also discuss any observations of signs of morbidity.

Item 2. Additional details should be provided in the Methods section pertaining to the quality of the virus stock, including mycoplasma status and genome copy numbers. Were ferrets positive for enteric coronavirus? The type of swab used for nasal samples should be indicated, i.e, pediatric or adult flocked swab, etc... Why were nasal virus loads assessed by swab as opposed to nasal wash?

Item 3. The transmitted viruses should be deep sequenced to examine potential adaptive signatures or quasi species expansion within the transmitted viral population.

Reviewer #2:

Remarks to the Author:

In the manuscript "SARS-CoV-2 is transmitted via contact and via the air between ferrets", the authors examine contact and airborne transmission of SARS-CoV-2 in 4 sets of animals intranasally inoculated with the virus. The authors demonstrate that the virus is efficient in transmission to all direct (4/4) and aerosol (3/4) contacts. In contrast to another report on this topic (ref 21), the authors detect infectious virus in the airborne recipient animals. This observation provides an important advancement to the field - that not only can airborne SARS-CoV-2 exposed animals seroconvert, but they become infected and shed significant amounts of virus for multiple days. These data provide a tractable model to ask important questions about intervention strategies to limit airborne transmission of SARS-CoV-2. There are a few details and modifications to the figures that the authors should consider in order to strengthen the conclusions, as well as provide important insight to compare transmission studies of SARS-CoV-2 that may emerge from multiple groups.

1. Please provide more details regarding the transmission setup, specifically the air flow rate in the cages, the room temperature/humidity, and the exposure duration of the direct and aerosol contacts. The last point is to clarify for this reviewer whether the exposure was for 21 days as loosely implied in the materials and method or 14 days as is standard for other transmission studies performed by this group. In addition, the authors may want to use these metrics, which are known to influence transmission of influenza viruses as a discussion point to contrast their results to those presented in ref 21 (Kim et al Cell Host and Microbe 2020).

2. It seems clear that 3 out of 4 airborne contacts shed virus in the upper respiratory tract (table 1). Viral titer data should be presented in Figure S1 or in the main text to accompany RNA titers, rather than a TCID50eq quantification. Especially given that the authors seem to have this data already and additional BSL3 experiments would not need to be performed.

3. Determination of antibody titers, presented in figure 2, does not seem to include endpoint titration.

The authors reference Ref 31 for a description of the ELISA methodology, but this manuscript (Okba et al Emerging Infectious Disease 2020) does not describe an ELISA protocol where serial dilutions of sera are used for endpoint quantification. The authors should consider doing a dilution series of the antibody to obtain the EC50 of antibody binding to Spike RBD. A neutralization would be preferred but to minimize additional BSL3 work, a sera dilution would be acceptable. Data on the day 0 serum should be included for each animal as well.

4. Please provide details on the swabs used for the nasal collection, as this is not common for collection of viruses from the ferret upper respiratory tract.

REVIEWER COMMENTS

Reviewer #1 (Remarks to the Author):

The manuscript submission by Sander Herfst and colleagues reports on the direct and airborne transmission of SARS-CoV-2 in a ferret model of COVID-19. The authors demonstrate in this animal model that virus transmission is higher by direct contact versus an airborne route of transmission. Their findings support infection prevention measures, such as social distancing, to limit the public health impact. There are several minor concerns that should be addressed prior to acceptance for publication.

Item 1. The data presented in supplementary figure 2 should be presented as figure 2 as infectious virus titers are more relevant for public health risk assessment. The manuscript should also discuss any observations of signs of morbidity.

The data presented in supplementary figure 2 also correspond to RNA quantification. Ct-values were extrapolated to TCID50eq/ml using a standard curve of a virus stock with a known infectious virus titer. However, to accommodate the comment of the reviewer, a supplementary figure (Supplementary Figure 3) displaying the infectious virus titers collected from the swabs of all ferrets was added.

The goal of the present experiment was to assess the transmissibility of SARS-CoV-2 between ferrets and not to assess its pathogenicity. Moreover, given the fact that ferrets are housed individually in the transmission set-up, it is very difficult to assess the presence of clinical signs. Therefore, we prefer not to describe any signs of morbidity in the present study.

Item 2. Additional details should be provided in the Methods section pertaining to the quality of the virus stock, including mycoplasma status and genome copy numbers. Were ferrets positive for enteric coronavirus? The type of swab used for nasal samples should be indicated, i.e, pediatric or adult flocked swab, etc... Why were nasal virus loads assessed by swab as opposed to nasal wash?

Details about the virus stock used to inoculate the animals were included in the Material and Methods. The ferrets were not tested for the ferret enteric coronavirus. However, given the fact that the ferret enteric coronavirus is an alphacoronavirus, it is very unlikely that antibodies against this virus would cross-react with SARS-CoV-2. The description of the type of swabs was added to the Material and Methods. Virus loads were assessed by nasal swabbing rather than nasal washing, as the latter might artificially create aerosols which could skew the results of the virus transmission via the air.

Item 3. The transmitted viruses should be deep sequenced to examine potential adaptive signatures or quasi species expansion within the transmitted viral population.

Whole genome consensus sequencing using MinION (Nanopore) was performed on throat swabs samples collected from donor, direct contact and indirect recipient ferrets. Mainly, two substitutions were detected in the spike protein in all ferrets: the N501T substitution, which is an ACE2 contact residue, and the S686G substitution, which is part of the furin cleavage site. In order to understand whether these substitutions were selected in ferrets from minority variants already present in the virus isolate or whether they emerge in ferrets, we performed Illumina next-generation sequencing on sequential samples from the donor ferrets and on the virus isolates. These data were added in a supplementary table (Supplementary Table 2) and described in the main text.

Reviewer #2 (Remarks to the Author):

In the manuscript "SARS-CoV-2 is transmitted via contact and via the air between ferrets", the authors examine contact and airborne transmission of SARS-CoV-2 in 4 sets of animals intranasally

inoculated with the virus. The authors demonstrate that the virus is efficient in transmission to all direct (4/4) and aerosol (3/4) contacts. In contrast to another report on this topic (ref 21), the authors detect infectious virus in the airborne recipient animals. This observation provides an important advancement to the field - that not only can airborne SARS-CoV-2 exposed animals seroconvert, but they become infected and shed significant amounts of virus for multiple days. These data provide a tractable model to ask important questions about intervention strategies to limit airborne transmission of SARS-CoV-2. There are a few details and modifications to the figures that the authors should consider in order to strengthen the conclusions, as well as provide important insight to compare transmission studies of SARS-CoV-2 that may emerge from multiple groups.

1. Please provide more details regarding the transmission setup, specifically the air flow rate in the cages, the room temperature/humidity, and the exposure duration of the direct and aerosol contacts. The last point is to clarify for this reviewer whether the exposure was for 21 days as loosely implied in the materials and method or 14 days as is standard for other transmission studies performed by this group. In addition, the authors may want to use these metrics, which are known to influence transmission of influenza viruses as a discussion point to contrast their results to those presented in ref 21 (Kim et al Cell Host and Microbe 2020).

Details specifying the air flow rate, the temperature of the room and exposure duration of direct contact and indirect recipient ferrets were added to the Material and Methods. We agree with the reviewer that using these metrics to compare our results with that of Kim et al would be interesting. However, we think that given the fact that many other parameters differ between the two experimental set-ups (distance between the ferrets, the number of ferrets per cage, the virus isolate and its passage history etc), it is difficult to really pinpoint those that would explain the observed differences without too much speculation.

2. It seems clear that 3 out of 4 airborne contacts shed virus in the upper respiratory tract (table 1). Viral titer data should be presented in Figure S1 or in the main text to accompany RNA titers, rather than a TCID₅₀eq quantification. Especially given that the authors seem to have this data already and additional BSL3 experiments would not need to be performed.

A supplementary figure (Supplementary Figure 3) displaying the virus infectious titers was added.

3. Determination of antibody titers, presented in figure 2, does not seem to include endpoint titration. The authors reference Ref 31 for a description of the ELISA methodology, but this manuscript (Okba et al Emerging Infectious Disease 2020) does not describe an ELISA protocol where serial dilutions of sera are used for endpoint quantification. The authors should consider doing a dilution series of the antibody to obtain the EC₅₀ of antibody binding to Spike RBD. A neutralization would be preferred but to minimize additional BSL3 work, a sera dilution would be acceptable. Data on the day 0 serum should be included for each animal as well.

A plaque reduction neutralization test was performed to determine the serum neutralization titer as the reciprocal of the highest dilution resulting in a reduction of >90% of the number of infected cells. These data were added as an additional panel to Figure 2. Sero-negativity for SARS-CoV-2 of the ferrets was already mentioned in the Material and Methods. For clarity, a sentence specifying that "All sera were tested negative by RBD ELISA and plaque reduction neutralization assay (OD₄₅₀ 0,02-0,05; PRNT <20)" was added to the legend of Figure 2.

4. Please provide details on the swabs used for the nasal collection, as this is not common for collection of viruses from the ferret upper respiratory tract.

Details on the swabs used for the nasal collection were included in the Material and Methods.

Reviewers' Comments:

Reviewer #1:

Remarks to the Author:

The authors have satisfactorily addressed the comments of the reviewer.

Reviewer #2:

Remarks to the Author:

In the revised manuscript "SARS-CoV-2 is transmitted via contact and via the air between ferrets", the authors examine contact and airborne transmission of SARS-CoV-2 in 4 sets of animals intranasally inoculated with the virus. The authors have addressed all of the previous concerns. In the revised manuscript the authors provide viral titer data and plaque reduction neutralization assays. The low viral infectious titer data presented in Supplemental Figure 3 for all animals (donor, direct contact, and indirect recipient animals) is surprising and the authors may want to comment on the distinction of RNA levels and infectious virus levels. Overall this work is very timely and will be of broad interest.

REVIEWERS' COMMENTS:

Reviewer #1 (Remarks to the Author):

The authors have satisfactorily addressed the comments of the reviewer.

Reviewer #2 (Remarks to the Author):

In the revised manuscript “SARS-CoV-2 is transmitted via contact and via the air between ferrets”, the authors examine contact and airborne transmission of SARS-CoV-2 in 4 sets of animals intranasally inoculated with the virus. The authors have addressed all of the previous concerns. In the revised manuscript the authors provide viral titer data and plaque reduction neutralization assays. The low viral infectious titer data presented in Supplemental Figure 3 for all animals (donor, direct contact, and indirect recipient animals) is surprising and the authors may want to comment on the distinction of RNA levels and infectious virus levels. Overall this work is very timely and will be of broad interest.

Low viral infectious titers are commonly observed in experimental models to study coronaviruses. Please see as an example other articles on SARS-CoV-2 in non-human primates (PMID: 32303590) and on MERS in rabbits (PMID: 31022948).